# Phospholipids: Identification and Implication in Muscle Pathophysiology

**DOI:** 10.3390/ijms22158176

**Published:** 2021-07-30

**Authors:** Rezlène Bargui, Audrey Solgadi, Bastien Prost, Mélanie Chester, Ana Ferreiro, Jérôme Piquereau, Maryline Moulin

**Affiliations:** 1Basic and Translational Myology Laboratory, CNRS UMR8251, Université de Paris, F-75013 Paris, France; rezleneh@yahoo.fr (R.B.); melaniechester19@gmail.com (M.C.); ana.b.ferreiro@gmail.com (A.F.); 2UMS-IPSIT-SAMM, Université Paris-Saclay, INSERM, CNRS, Ingénierie et Plateformes au Service de l’Innovation Thérapeutique, F-92296 Châtenay-Malabry, France; audrey.solgadi@universite-paris-saclay.fr (A.S.); bastien.prost@universite-paris-saclay.fr (B.P.); 3AP-HP, Reference Centre for Neuromuscular Disorders, Institute of Myology, Pitié-Salpêtrière Hospital, F-75013 Paris, France; 4Signalling and Cardiovascular Pathophysiology, INSERM UMR1180, Université Paris-Saclay, F-92296 Châtenay-Malabry, France; jerome.piquereau@universite-paris-saclay.fr

**Keywords:** phospholipids, mass spectrometry, muscle, disease, membrane, endoplasmic reticulum, mitochondria

## Abstract

Phospholipids (PLs) are amphiphilic molecules that were essential for life to become cellular. PLs have not only a key role in compartmentation as they are the main components of membrane, but they are also involved in cell signaling, cell metabolism, and even cell pathophysiology. Considered for a long time to simply be structural elements of membranes, phospholipids are increasingly being viewed as sensors of their environment and regulators of many metabolic processes. After presenting their main characteristics, we expose the increasing methods of PL detection and identification that help to understand their key role in life processes. Interest and importance of PL homeostasis is growing as pathogenic variants in genes involved in PL biosynthesis and/or remodeling are linked to human diseases. We here review diseases that involve deregulation of PL homeostasis and present a predominantly muscular phenotype.

## 1. Introduction

Life is amazing. The organization of ‘only’ four major types of biological macromolecules constitutes the majority of the dry weight of the human body. Specifically, carbohydrates, proteins, nucleic acids, and lipids perform an incredible variety of functions in the organism. Lipids, also previously known as ‘lipin’ and ‘lipoid’, are a huge family without a real consensual definition from a chemical or biological point of view. In 1920, Bloor from the department of Biochemistry and Pharmacology of The University of California, proposed a classification of lipids in three groups, simple lipids with fats, oils and waxes; compound lipids including phospholipids (PLs) and glycolipids; and a last group including derived lipids (fatty acids (FA), steroids, etc.) [1]. Of all the different lipids, PLs were indispensable to the appearance of life since they can delimit cellular compartments [2,3]. After presentation of the basic concepts on PLs, we present how to isolate, separate, and identify these molecules. The importance of PLs in numerous biological functions is highlighted by rare genetic diseases that involved deregulation of PLs homeostasis. We will center this review on diseases with a predominantly muscular phenotype.

## 2. Basic Concepts on PLs

Phospholipids include phosphatidic acid (PA), phosphatidylglycerol (PG), phosphatiylcholine (PC), phosphatidylethanolamine (PE), phosphatidylinositol (PI), phosphatidylserine (PS), cardiolipin (CL), and sphingomyelin (SM). Among the PLs, the most abundant are glycerophospholipids (or phosphoglycerides), whose common structure comprises a polar head containing a glycerol to which is attached a phosphate group carrying an additional molecule (which gives its identity to the lipid) and two fatty acids (hydrophobic hydrocarbon chains, sn1 and sn2) [4] (cf. Figure 1). The polar head group can be simply a hydrogene in the case of phosphatidic acid; be an ethanolamine, choline, serine, inositol, or glycerol group; and even a phosphatidylglycerol for cardiolipin. Another abundant type of PL is represented by sphingomyelins, which are phosphosphingolipids (a sub-class of sphingolipids). SM contains a sphingosine (a fatty alcohol containing 18 carbons) bound to phosphocholine and a fatty acid. PLs are specialized lipids, which have multiple names depending on biologist or chemist point of view despite a well-defined but diverse and complex structure. Several classifications of lipids exist according to, for example, their chemical structure or biological origin, but without any real consensus [4]. For example, the fatty acid names depend on the carbon length and the presence of double bonds, but the numbering can be different from the biological or the chemical point of view (cf. Figure 2 and Table 1). Nevertheless, the International Union of Pure and Applied Chemistry (IUPAC) has brought about progress into establishing a chemical nomenclature and terminology for specific scientific fields (https://iupac.org/, accessed on 3 May 2021). Moreover, the database LIPID MAPS (Metabolites and Pathways Strategy—https://www.lipidsmaps.org/, accessed on 3 May 2021) constitutes an important tool for lipidomic analysis, with structures and annotations of bulk lipids [5,6,7]. Importantly, most organic molecules are largely mobile depending on their physical and chemical environment (cf. Figure 2).

The molecular diversity of phospholipids is very large and depends not only on the polar head, the carbon length (short/medium/long/very long), and the number of carbon chains (from one in lyso-PL to four in cardiolipin), but also on the degree of double bonds (responsible for molecular rigidity) and their position (cf. Figure 2). There are also plasmalogens, PL analogs containing an ether or vinyl ether group instead of the usual ester function [8]. The position of a fatty acyl chain in sn1 or sn2 increases the diversity further. Thus, the beautiful and large family of phospholipids already contains more than 40,000 natural molecules [2,9]. Precise and structural identification of lipids in a biological medium has been underway for more than 30 years and is becoming possible thanks to the development of mass spectrometry [10]. Of note, the use of common name of fatty acid should be carefully used unless the position of the double bond and the cis and trans isomeres are fully identified. We advise using the systematic name if no information on double bond position is provided (cf. Table 1).

**Figure 2 ijms-22-08176-f002:**
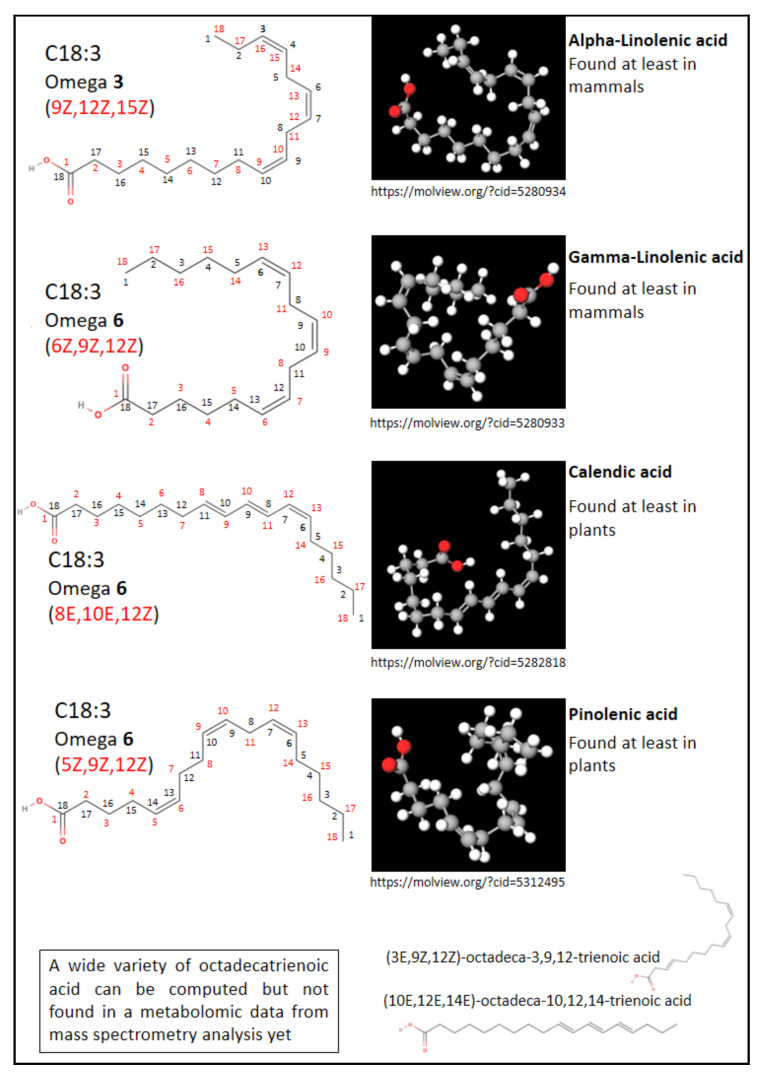
FA nomenclature. The chemical structure of FA showing the physiological numbering (black), the first double bond being at the third or sixth carbon from the omega start, and the chemical numbering (red) with conventions for the double bond location (E in cis, Z in trans). One of numerous possibilities of 3D representation was obtained thanks to Molview [11]. 3D structure is influenced by local environments such as pH.

In order to present the different PLs in more detail, we chose to follow the order of lipid retention following liquid chromatography using a polar stationary phase (polyvinyl alcohol grafted silica column). This allows phospholipids to be separated by polarity, followed by a charged aerosol detector Corona CAD and a high-resolution mass spectrometer [12] (cf. Figure 3 and Table 2). Phosphatidic acid is the simplest PL, one of the least abundant PL present in the plasma membrane, but is central in PL biosynthesis [13]. PA has been involved mainly in membrane biophysical properties and in various signaling pathways such as adipogenesis or autophagy [14]. Three major pathways can generate PA: (i) From the acylation of lyso-PA (LPA) by lysophosphatidic acid acyltransferases (LPAAT) (cf. Figure 4). LPA can be obtained from glycerol-3-phosphate (G3P) thanks to the enzymatic action of glycerol-3-phosphate acyltransferase (GPAT). (ii) From the hydrolysis of phosphatidylcholine by the phospholipase D. (iii) From the phosphorylation of diacylglycerol (DAG) by a diacylglycerol kinase (DGK 10 isoforms in human) [13,15]. Importantly, PA is a major precursor of PL biosynthesis. DAG and CDP-DAG (cytidine diphosphate-DAG) are key intermediates that can be obtained from PA thanks to phosphatidic acid phosphatases (PAP or diacylglycerol-3-phosphate phosphohydrolase as systematic name) and CDP-DAG synthetase (CDS or CTP:phosphatidate cytidylyltransferase as systematic name), respectively [14]. They are two types of PAP, type I encoded by lipin genes and type II mainly found in the plasma membrane known as lipid phosphate phosphatases.

From the liponucleotide CDP-DAG, phosphatidylglycerol and 1,3-bis(sn-3′-phosphatidyl)-sn-glycerol known as cardiolipin and phosphatidylinositol can be synthetized [16]. PG has low abundance, less than 1% of total PLs in mammal cells, and is relatively saturated compared to other PLs [17]. Cardiolipins are unique among the PL family as they are specifically present in mitochondria membranes, especially in inner membrane, and composed of a dimeric phosphatidic acid containing two phosphates and four acyl chains [18] with molecular structure similarity with PG [16,19]. This distinctive four acyl chain composition induces a cone-shaped PL that has specific properties such negative membrane curvature and proton buffering [20]. Moreover, this specific structure leads to an important variety of CL species due to a huge diversity of acyl composition. From the biosynthesized ‘nascent’ CL, there is further remodeling leading to hetero-acylated or homo-acylated CL [16]. Three different enzymes are involved in acyl synthetic transformation: tafazzin (a trans-acylase) and two acyl-transferases MLCAT1 (monolysoCL acyltransferase 1) and ALCAT1 (acylCoA:lysoCL acyltransferase 1) [16]. This lipid dimer with four acyl chains was initially isolated from heart beef as cardiomyocytes are the richest cells in terms of mitochondria [21]. In heart and skeletal muscle, the main cardiolipin specie is composed of four C18:2 acyl chains. The last PL synthesized from CDP-DAG is PI, and the biosynthesis takes place in the ER thanks to PIS (phosphatidylinositol synthase), whereas PG and CL biosynthesis are orchestrated in mitochondria [18]. Despite a relatively minor representation in membrane composition (around 10%), PI is an important PL due to the generation of several phosphorylated derivatives at the inositol ring, which are involved in a plethora of cell signalizations such as membrane trafficking and regulation of protein function [18]. Strikingly, the fatty acyl composition of PI is mainly C18:0/C20:4 in numerous rodent tissues, and the consequences of this specific fatty acyl composition are not well understood [22].

PC and PE, the two most abundant phospholipids, are generated by the Kennedy pathway by which DAG is transferred to CDP-choline and CDP-ethanolamine, respectively (cf. Figure 4). These intermediates are obtained from phospho-choline and phospho-ethanolamine thanks to CTP:phosphocholine cytidylyltransferase (CT) and CTP:phosphoethanolamine PE cytidylyltransferase (ET) [23]. PE can generate PC by methylation reactions by phosphatidylethanolamine N-methyltransferase (PEMT). PE can also be produced specifically in the mitochondria by a phosphatidylserine decarboxylase (PSD) from PS. Two enzyme phosphatidylserine synthase 1 and 2 responsible for base-exchange reactions synthesize PS from PC and PE, respectively [24]. PC are also involved in sphingomyelins (which do not have glycerol in their structure) synthesis in the Golgi from ceramides (produced in the ER) by sphingomyelin synthase (SMS), which transfers phosphocholine from a PC [25].

Details in PL biosynthesis pathways are reviewed in [19,26,27]. Multiple pathways can lead to PL synthesis, re-synthesis, or remodeling using various intermediates, and these pathways could happen in different membranes such as plasma membrane, ER membrane, lysosome/peroxisome membrane, or mitochondria membranes. De-acylation and re-acylation reactions due to a large number of phospholipase enzymes add further difficulties in pinpoint the precise PL composition [28]. Each cell has a unique PL fingerprint, making it difficult to unravel the physiological and pathophysiological significance [18]. The development of methods that track PL movement from their place of synthesis to their final organelle destination will help to decipher their role in cells.

**Table 2 ijms-22-08176-t002:** Main characteristics of phospholipids.

Phospholipid Group	% of Total Lipids in Eukaryotic Cells [27]	Main Biosynthesis Site	Main Characteristics
Phosphatidic acid PA	1–2	ER	The simplest PL Main precursor for PL synthesis Inverse cone
Phosphatidylglycerol PG	<1	Mitochondria	Precursor of cardiolipin
Cardiolipin CL	2–5	Mitochondria	Present only in mitochondrial membranes Contains four fatty acyl chains Inverse cone
Phosphatidylinositol PI	10–15	ER	Source of inositol 1,4,5 triphosphate (IP3) Cylinder shape
Phosphatidylethanolamine PE	15–25	ER and also mitochondria from PS	Second most abundant PL Inverse cone
Phosphatidylserine PS	5–10	ER	Mostly present in the inner leaflet of the cell membrane, when externalized they became a signal of recognition for cell phagocytosis by macrophages (characteristic of apoptosis) Cylinder shape
Lysophosphatidylethanolamine LPE	ND	Obtained from phospholipase action	Contains one fatty acyl chain Hydrolysis of one fatty acid of a PE by a phospholipase
Phosphatidylcholine PC	45–55	ER	The most abundant (outer leaflet of cell membrane) Constitutes a reserve of choline and methyl groups Formerly called lecithin Cylinder shape
Sphingomyelin SM	5–10	Golgi	Phosphocholine bound to ceramide Abundant on the outer leaflet of the membrane Cylinder shape
Lysophosphatidylcholine LPC	ND	Obtained from phospholipase action	Contains one fatty acyl chain Hydrolysis of one fatty acid of a PC by a phospholipase

## 3. Main Methods for PL Analysis

In 2021, more than 200,000 results were obtained through a PubMed search using the keyword “phospholipids”. Evolution of the number of publications is closely linked to technological progress in the lipid research field, particularly with chromatography techniques (first thin layer, then gas and liquid phase) associated with ever more efficient detectors. Since the early 2000s, metabolomic screening approaches using coupled mass spectrometry have been flourishing.

Liquid–liquid extraction (LLE) is the most widely used method for extracting lipids from a biological medium. The first LLE method was developed by Folch in 1957 using a mixture of chloroform, methanol, and water (8:4:3 *v*/*v*/*v*) [29]. Most of the lipids (from a wide range of polarity) are recovered from the sample [30]. However, PL extraction yield can vary from 50 to 90%, some of them remaining in the aqueous phase [31]. The Folch method was subsequently modified by Bligh and Dyer to increase the proportion of polar lipids extracted by using a 1:2:0.8 *v*/*v*/*v* mixture of chloroform, methanol, and water [32]. Since then, many protocols have been developed to optimize extraction efficiency according to the lipids of interest or the nature of biological samples studied and to limit the use of toxic solvents such as chloroform [33].

The characterization of PL molecular species is mainly performed with mass spectrometry (MS) tools generally coupled with chromatographic techniques. Various separation methods have been developed, depending on the lipid mixture and the questions to answer [34,35]. Gas chromatography (GC), often used for fatty acid analysis, is not suitable for whole PL analysis as PLs are too polar and not very volatile. The overall FA composition can be obtained by GC/MS after hydrolysis of PLs, but this method does not allow for identification of plasmalogens ether or vinyl ether acyl chain. Liquid chromatography (LC) is much more suitable for PL analysis including plasmalogens. Normal phase HPLC (high-performance LC) has the advantage of separating PLs according to the polar head nature (allowing quantification of PL classes on a reduced retention time). This facilitates the structural identification of molecular species; however, it requires the use of hazardous solvents [36]. Reverse phase HPLC separates PL molecular species on the basis of hydrophobicity of FA chains, depending on carbon length and number of unsaturations; this induces mixing of lipid classes and difficulties for the structural characterization of species [37]. A more environmentally friendly approach is the super critical fluid chromatography (SFC), which has the advantages of normal phase HPLC while using eco-compatible (CO_2_ in super critical phase, H_2_O) and less toxic solvents (acetonitrile) [38].

Once they are separated by chromatography, mass spectrometry is the most efficient tool for characterizing PLs, which ionize very well in electrospray. The analysis can be done in a non-targeted (profiling) or targeted (semi-quantitative) way [39]. In lipidomic profiling, high-resolution mass spectrometry allows to obtain the atomic composition of the whole molecule (generally in adduct form). If chromatography does not provide information on PL classes, fragmentation spectra are essential to identify the molecular structure precisely. Positive mode fragmentation confirms the nature of the polar head, while negative mode fragmentation gives access to FA composition [40]. Although MS is a powerful tool for structural characterization, it provides limited information on positioning of unsaturation on FA chains [38]. Specific methods have been developed with this aim, using alkaline adducts or ozone as reactant gas combined with targeted fragmentation. However, implementation of theses analyses and spectra interpretation are still highly challenging [35]. In the same way, assignment of sn-positions of PL acyl chains is only possible through validated assays based on monitoring the ratio between fragment ions of positional isomers.

Mass spectrometry coupled with chromatographic techniques are lengthy forms of analysis (several dozen minutes) and consume important volumes of solvents. MS can also be used alone, without prior separation, in shotgun MS analysis [41]. This technique is fast and simple to implement, giving access to a global phospholipidomic profile. Very high spectral resolution is required to identify molecular species with similar masses, and separation of isomers is not possible with shotgun analysis. Simultaneous ionization of hundreds of molecules, with different concentrations and polarities leads to ionization suppression, limiting the visibility of low-concentration and/or poorly ionizable species [42].

Quantification of lipid species remains challenging as molecular standards of each species are not available [38]. FA composition will strongly influence ionization, making analogy with different molecular species not possible [43]. In general, the quantification of PLs by species is semi-quantitative.

Phospholipidomics provides large amount of valuable information that implies statistical analysis like other ‘omics’ approaches. Large-scale profiling gives access to the phospholipidome of cells, tissues, or various samples at a specific time. This PL fingerprint is an important step towards understanding PL metabolism and disturbances.

## 4. Phospholipids in Biological Membranes

The fluid-mosaic membrane model was proposed from the 1970s by Singer and Nicolson and revisited in 2014 by Nicolson [44,45]. Membranes, interacting with the cytoskeletal network and extracellular matrix, are dynamic structures composed not only of lipids, but also of membrane proteins. Dynamic and reversible micro and even nano-size structures, such as lipid rafts, can be present in the biological membranes [45,46,47]. Membranes are integrated elements within cells, tissues, and organisms; they are sensitive and can react to variation of their environment [45]. We will focus on PLs, the main lipid present in the asymmetrical cell membranes [46]. Their inner leaflet is composed of PS, PA, and PI, which induce a strong negative charge compared to the relatively uncharged outer membrane with a majority of PC and SM [48]. As mentioned previously, the main site of phospholipid biosynthesis is the endoplasmic reticulum membrane, followed by mitochondrial membranes; these are two key organelles for cellular homeostasis, controlling among other things calcium content and ATP production [27]. Vance’s research has shown in the beginning of the 1990s that ‘membrane bridges’ between the ER and mitochondria were able to synthesize a large variety of phospholipids [49,50]. These two organelles are in some places very close (around 10 nm apart), and these connecting zones have been named mitochondria-associated membranes (MAM) or mitochondria–ER connect (MERC) [51,52,53,54]. The MAMs allow for the existence of calcium microdomains [55] and energy microdomains [56], which require proximity between mitochondria and ER, but also exchange of PL, although the transporters associated with certain PL have not been clearly identified [52,54]. The distribution of PL species varies among the different organelles [27,57] and can evolve with changes in the cellular environment and with events such as aging, diet, or the circadian clock [58,59,60].

Membranes are dynamic structures. This is especially well known for mitochondria as they can form a filamentous or a fragmented network [59,61,62]. The inner leaflet of mitochondria is very peculiar since it includes only three major PLs: nearly 40% PC, 40% PE, and more than 15% cardiolipin. This composition, rich in PE of conical shape and CL (four fatty acid chains), allows for the formation of numerous invaginations of the inner membrane, the mitochondrial cristae. Regarding the ER, more than half of its PLs are PCs. The ER is composed of domains that perform multiple functions, leading to a heterogeneous protein distribution. A distinction is made between the smooth ribosome-free ER and the rough ribosome-containing ER, which is involved in protein biosynthesis. Muscle cells possess not only the classic ER but also a vast network of specialized membranes associated with myofibrils called the sarcoplasmic reticulum (SR) [51]. The SR allows the intake, storage, and flow of calcium required for muscle contraction. Avoiding calcium leakage from the ER/SR and uncoupling of mitochondria is essential for optimal cellular function. The integrity of the membranes of these organelles is crucial for this, and phospholipids should play an important role [63].

Another important parameter in membrane composition is cholesterol, particularly for plasma membranes [46]. As the molecular ratio is roughly 0.8 cholesterol per phospholipid in plasma membranes [48], their interaction is surely a key parameter for cell homeostasis, cell signaling, and pathophysiology. Two of the key partners for cholesterol are PC and sphingomyelins [45], which are mainly present in the plasma membrane and lysosomes.

The PL composition of the membranes of the numerous organelles and the plasma membrane is adapted to each cell type to meet different physiological needs [64,65]. Depending on the type of carbon chains of each PL, the membranes will be more or less fluid, flexible, compact, and thick, giving them particular properties [64]. The PLs with saturated fatty acids make the membranes rigid, whereas the membranes that contain fatty acids with several unsaturations will be more fluid [66]. The state (fluid vs. rigid) of the membrane can modify cellular processes such as membrane traffic, vesicle fusion, or formation of lipo-protein complexes at the membrane [64].

## 5. Importance of PLs in Striated Muscle Functions

The local coupling of the sarco-endoplasmic reticulum and mitochondria, which controls muscle contraction and energy production, appears central to skeletal and cardiac muscle function and dysfunction. Integrity of the membrane organelles is crucial to limit calcium leak from the ER/SR and mitochondrial proton and electron escape. Furthermore, skeletal muscles are subjected to an aperiodic and highly variable mechanical force that is transmitted to membranes [67]. Thus, it is important to study more in detail the membrane composition, particularly the amphiphilic molecules that participate in optimal muscle function. Many questions remain partially unresolved; for example, how are they essential to the dialogue between ER and mitochondria? What links exist between phospholipids and skeletal or cardiac muscle diseases? Is PLs dyshomeostasis a cause or a consequence of muscle dysfunction?

Indeed, PL imbalances can disrupt cell and organelle homeostasis and participate in various pathologies [68]. The indisputable evidence of these points comes from rare genetic diseases. In the Orphanet classification, a group of disorders of PLs, sphingolipids, and fatty acids biosynthesis (ORPHA:352301) was created in 2013 [69,70] and is divided in three sub-groups depending on the predominant phenotypical involvement, namely, (i) central nervous system (ORPHA:352306), (ii) peripheral nerves (ORPHA:352309), and (iii) skeletal muscle (ORPHA:352312). The latter highlights a key role of PLs in muscle physiology with five genetic diseases in which the primary defect directly targets PL: Barth syndrome (ORPHA:111, associated with defects of the *TAZ* gene encoding tafazzin), congenital cataract-hypertrophic cardiomyopathy-mitochondrial myopathy syndrome also known as Sengers syndrome (ORPHA:1369, due to defects of *AGK* acylglycerol kinase, SLC25A4 solute carrier family 25 member 4), genetic recurrent myoglobinuria (ORPHA:99845, associated with the Lipin1 gene *LPIN1*, *MTCO1*, or *MTCO2* mitochondria DNA-encoded cytochrome C oxidase), megaconial congenital muscular dystrophy (ORPHA:280671—*CHKB*, choline kinase beta), and neutral lipid storage disease (ORPHA:165—*ABHD5*, abhydrolase domain containing 5, *PNPLA2/ATGL* patatin-like phospholipase domain-containing 2) (cf. Table 3).

Pathogenic variants in the *TAZ* gene, encoding a transacylase necessary for cardiolipin remodeling, are responsible for Barth’s syndrome (BTHS), a devastating disease affecting the neuromuscular and metabolic systems, characterized by dilated cardiomyopathy and skeletal myopathy in infants [128]. Despite normal content of PE and PC, decreased levels of long polyunsaturated chains (20:4 and 22:6) and increased of 18:2 (probably linoleic acid) has been described in children with this syndrome [129]. Mitochondria in BTHS have important alterations of the cristae ultrastructure (fold and invagination of the inner mitochondrial membrane) leading to defective ATP production [130,131]. Studies involving patients with Barth syndrome are extremely valuable to highlight the key roles played by CLs in mitochondrial respiratory function. Thus, the alteration of CL maturation in these patients has been shown to lead to an instability of respiratory chain supercomplexes that affects complex I activity, thereby demonstrating the importance of CLs for complex I-containing supercomplex formation and function [132].

Sengers syndrome, due to pathogenic variants in the *AGK* gene, shares strong similarities with Barth syndrome, including skeletal myopathy and cardiomyopathy (hypertrophic), being lethal within the first year of life in around half of the patients [133]. The AGK kinase is involved in phosphorylation of MAG (monoacylglycerol) to LPA and DAG to PA, which are largely involved in PL biosynthesis. The depletion by siRNA of AGK in PC-3 prostate cancer cells induce a reduction of LPA and PA in mitochondria [134], whereas in HEK293 (human embryonic kidney), no changes were observed [135]. To the best of our knowledge, no phospholipidomic data are available in muscle cells. Moreover, the protein belongs to the translocase of the inner mitochondrial membrane TIM22 complex [136]. This multi-protein complex is involved in the import and assembly of mitochondrial carrier proteins such as ANT (adenine nucleotide translocator) [136].

Pathogenic variants in the *CHKB* gene, encoding choline kinase beta, an enzyme involved in the synthesis of phosphatidylcholine, cause megaconial congenital muscular dystrophy in humans (associated with developmental delay and autistic behavior) and to rostrocaudal muscular dystrophy in mouse [86,87,88]. Decreased content of this most abundant phospholipid in the eukaryotic membrane leads to giant mitochondria localized in the periphery of muscle fibers [137]. Mitochondrial membrane potential is decreased in patient-derived cells, and a reduction of mitochondrial respiratory complex I activity is observed in their skeletal muscles [138].

Patients with a genetic defect causing loss of function of *LPIN1* (gene coding for LIPIN 1) develop myopathy, affecting both skeletal muscle and the heart [81]. Investigations into the links between lipid balance, more specifically phospholipid balance, and skeletal muscle function have confirmed that modification of certain proteins involved in the biosynthesis of PLs influences PLs levels. Loss of *Lpin1* in mice alters PL levels in gastrocnemius muscle, with significant increase in PC, PE, PI, PS, and PG levels, as well as in PA levels. LIPIN 1, a Mg^2+^-dependent phosphatidate phosphatase, thus appears to be an important enzyme in the homeostasis of phospholipids in muscle [81].

Other genes involved in PL, sphingolipid, and fatty acid biosynthesis leading to a predominantly muscular phenotype when mutated in humans could extend the ORPHA:352312 group (cf. Table 3). For example, pathogenic variants of *HACD1* (3-hydroxyacyl-CoA dehydratase 1), an ER resident enzyme involved in the synthesis of very long-chain fatty acids, have been associated with congenital myopathies in humans and in dogs [90,91,139]. In Hacd1-deficient mice, skeletal muscle mitochondria exhibit a significant decrease in CL content in the mitochondrial inner membrane [91]. This is associated with important cristae remodeling and strong alterations of mitochondrial coupling, leading to a higher energy dissipation. An in vitro assay showed that enrichment of mitochondrial membranes with CLs rescued the coupling efficiency of ATP synthase and therefore the mitochondrial electron transfer chain function, highlighting the key role of CLs in the respiratory chain [117,121,122,123].

Alteration of PLs is also present in inherited form of myopathies due to pathogenic variant in genes without a direct link to PL biosynthesis (cf. Table 3). In muscle from *DMD* (Duchenne muscular dystrophy) patients, increased PC and SM have been reported [119]. In EDL muscle from the mdx mice (the murine model of DMD, devoid of dystrophin), an increase PC containing saturated acyl chain (16:0 and 18:0) and a decrease PE and PC containing long and polyunsaturated acyl chain (22:6) and (20:4) were found. Genetic rescue in mdx mice leading to truncated dystrophin expression was able to restore the PL profiles [121]. Additionally, the increased PC/PE ratio in mdx mice is corrected by the transgenic expression of FAS (fatty acid synthase), stearoyl-CoA desaturase-1 (SCD1), and Lipin1 [140]. In dysferlinopathies, muscle diseases due to pathogenic variants in the gene encoding dysferlin, a transmembrane protein implicated in protein trafficking, lipid droplets accumulate in myofibers [141]. Molecular remodeling of the lipidome has been observed in skeletal muscle of BLAJ mouse (a dysferlinopathy model), with increased content of species of sphingomyelin, PC, LPC, PE, and LPE in quadriceps from BLAJ compared to WT [142].

In skeletal muscle, ER stress and the stress response provided by the UPR (unfolding protein response) system can be activated under different conditions, including inherited myopathies. The UPR system may be triggered by the accumulation of misfolded proteins, altered calcium balance, or the presence of reactive oxygen species (ROS) [143,144]. Since PLs are synthesized in the ER and constitute its membranes, a link can exist between alteration of the phospholipid balance and reticulum damage. Indeed, loss of *Lpin1* gene function in mice induced, in addition to altered PL composition, a significant increase of different ER stress markers (Bip, Fgf21, Gdf15, HSP90b1, and Edem1) [81]. This study also showed morphological alteration of the mitochondria associated with ER stress [81], consistent with the fact that the maintenance of the phospholipid balance is also ensured by the exchange of PLs between ER and mitochondria via the MAMs, and that mitochondria are a site for the synthesis of certain PLs such as PG or CL.

Thus, PL alterations in mouse muscles can be associated with changes in other pathways, such as ER stress, mitochondrial dysfunction, and disturbed cell homeostasis. Taken together, these changes are associated with different inherited myopathies. Although all the roles of PLs at the muscle level are not yet clearly defined, these molecules seem to be important to maintain cell homeostasis and organelle integrity, ensuring normal muscle function. Therefore, a so-called ‘normal’ composition of PLs or PL homeostasis, particularly in the ER/RS membranes and mitochondria, key structures in muscle, is important for maintain motor function, contraction, muscle development and regeneration.

## 6. Conclusions

Phospholipids are a large family of diverse lipids with a wide variety of structures and physico-chemical properties. The role of PLs in the physiology and pathophysiology of skeletal muscles has been underanalyzed, although muscle is one of the largest metabolic organs. More efforts should be made to fully understand the mechanisms of PL dyshomeostasis and their pathophysiological relevance, which may further improve our understanding of muscular diseases. In particular, characterizing the mechanisms by which PLs modulate the endoplasmic reticulum and mitochondria functions could contribute to a better understanding of the pathophysiology of inherited or acquired myopathies. Thanks to the development of new generation of advanced lipid analysis technologies, phospholipidomics can now help elucidate the importance of PLs in muscle function. Comprehensive PL profiling is necessary for the discovery of diagnostic biomarkers and new therapeutic targets, as well as for the development of personalized therapeutic approaches to treat or slow the progression of muscle pathologies.

## Figures and Tables

**Figure 1 ijms-22-08176-f001:**
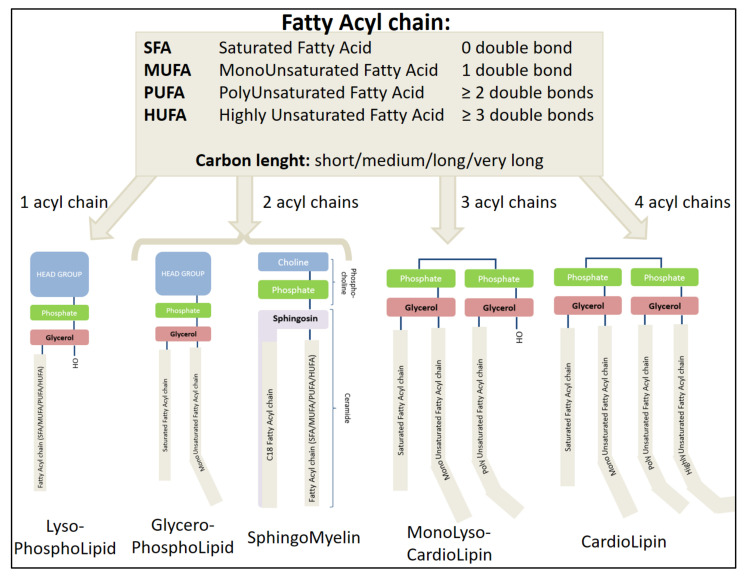
The great diversity of phospholipids.

**Figure 3 ijms-22-08176-f003:**
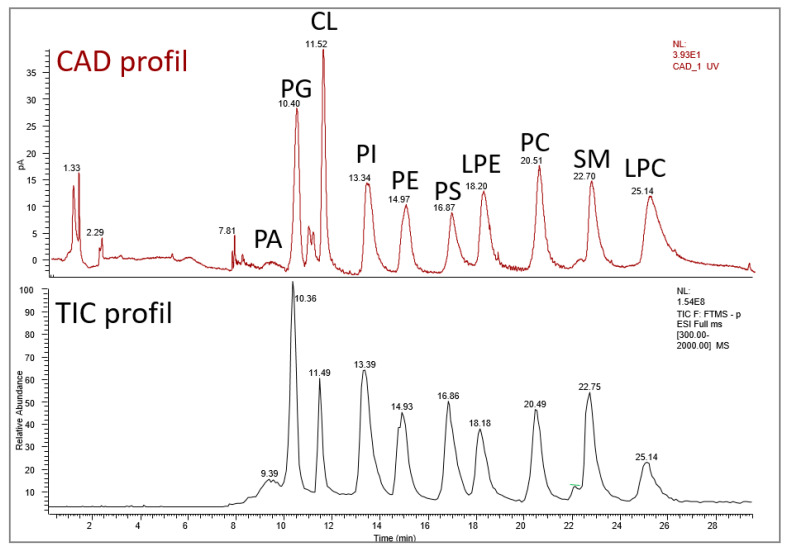
TIC (total ion chromatogram) by MS ESI-neg profiling and Corona CAD analysis for phospholipid class detection. Standard analysis of 0.3 g/L of PA, PG, CL, PI, PE, PS, LPE, PC, SM, and LPC.

**Figure 4 ijms-22-08176-f004:**
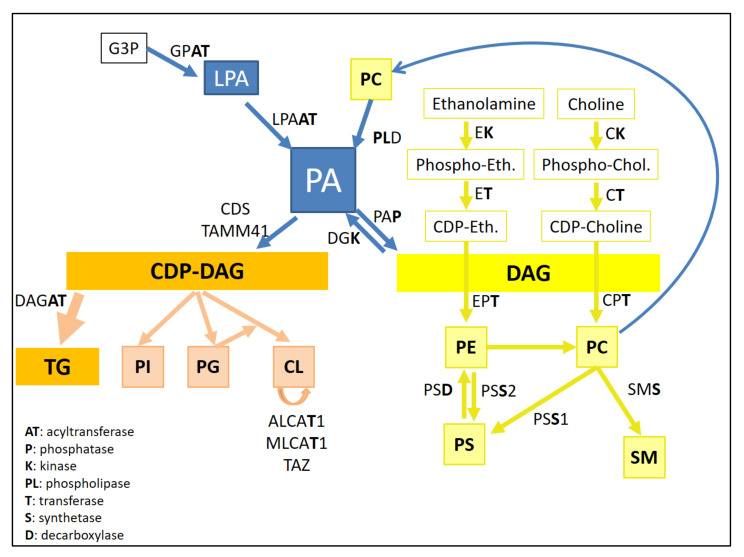
Diacylglycerol (DAG) and CDP-DAG are two key elements in PL biosynthesis. GPAT: glycerol-3-phosphate acyltransferase, LPAAT: lysophosphatidic acid acyltransferases, PAP: phosphatidic acid phosphatases, DGK: diacylglycerol kinase, CDS: CDP-DAG synthetase, DAGAT: diacylglycerol acyl transferase, ALCAT1: AcylCoA:lysoCL acyltransferase 1, MLCAT1: monolysoCL acyltransferase 1, TAZ: tafazzin, EK: ethanolamine kinase, ET: CTP:phosphoethanolamine PE cytidylyltransferase, EPT: ethanolaminephosphotransferase, PSD: phosphatidylserine decarboxylase, PSS2: phosphatidylserine synthase 2, CK: choline kinase, CT: CTP: phosphocholine cytidylyltransferase, CPT: cholinephosphotransferase, PSS1: phosphatidylserine synthase 1, SMS: sphingomyelin synthase.

**Table 1 ijms-22-08176-t001:** The importance of the double bond position.

Fatty Acid	18:1			18:2			18:3		
**Mass and** **Formula**	282.26 and C_18_H_34_O_2_			280.24 and C_18_H_32_O_2_			278.22 and C_18_H_30_O_2_		
**Systemic Name**	Octadecenoic acid			Octadecadienoic acid			Octadecatrienoic acid		
**Number of** **Lipid Records**	43			101			63		
	**Common Name**	**Double Bond** **Position**	**Number of** **PubMed** **References**	**Common Name**	**Double Bond** **Position**	**Number of PubMed** **References**	**Common Name**	**Double Bond** **Position**	**Number of** **PubMed** **References**
**Example**	Oleic acid	9Z	23,694	Linoleic acid	9Z, 12Z	24,310	α-Linolenic acid	9Z, 12Z, 15Z	7036
	Trans- vaccenic acid	11E	325	Rumenic acid	9Z, 11E	316	γ-Linolenic acid	6Z, 9Z, 12Z	3145
	Cis-vaccenic acid	11Z	202	Linoelaidic acid	9E, 12E	25	Punicic acid	9Z, 11E, 13Z	111
	Petroselinic acid	6Z	67	Sebaleic acid	5Z, 8Z	9	Pinolenic acid	5Z, 9Z, 12Z	56
	Petroselaidic acid	6E	4	Vaccelenic acid	11E, 15Z	2	Jacaric acid	8Z, 10E, 12Z	13

Formula of the fatty acid can be deduced from its systematic name. Records lipid from https://lipidmaps.org/data/chemdb_lm_text_ontology.php?ABBREV=FA%2018:3 (accessed on 3 May 2021) and records from PubMed https://pubmed.ncbi.nlm.nih.gov/ (accessed on 3 May 2021). (E: trans double bond, Z: cis double bond).

**Table 3 ijms-22-08176-t003:** PL alterations involved in muscular diseases.

	Gene Mutated	Protein and Function	Disease	Muscular Symptoms	Lipid Disorders	Animal Experimental Models	Ref.
ORPHA 352312: group of disorders of PL, sphingolipids and FA biosynthesis with skeletal muscle characteristics	*TAZ* chromosome X	Tafazzin Phospholipid-lysophospholipid transacylase Catalyzes remodeling of immature CL to its mature composition	Barth syndrome	Dilated cardiomyopathy Skeletal muscle weakness	Reduced level of CL Increased level of mono-lysocardiolipin	*Taz^−/−^* (knock-out) mice: most die before birth due to skeletal muscle weakness, survivors develop progressive cardiomyopathy *Taz* inducible knock-down mice (doxycycline-inducible short hairpin RNA): impact on cognitive abilities, brain mitochondrial respiration and the function of hippocampal neurons and glia *Tafazzin* mutant Drosophila melanogaster: motor weakness (reduced flying and climbing abilities)	[71,72,73]
	*AGK* chromosome 7	AGK Acylglycerol kinase Phosphorylates MAG and DAG to form LPA and PA	Sengers syndrome	Hypertrophic cardiomyopathy Skeletal myopathy Congenital cataract	Defects in mitochondria and storage of lipids in muscle (skeletal and heart)	*Agk*^−/−^ mice: phenotype similar to Sengers syndrome, develop thrombocytopenia and splenomegaly	[74,75,76,77]
	*LPIN 1* chromosome 2	Lipin-1 Phosphatidate phosphatase Catalyzes the conversion of PA to DAG	Myoglobinuria	Rhabdomyolysis Muscle fiber loss	Hyper-triglyceridemia Accumulation of PL intermediates	Spontaneous mutant mice ‘*fld*’ (fatty liver dystrophy): neonatal mice: hypertriglyceridemia andenlarged, triglyceride-rich fatty liver after 2 weeks: neuropathy with abnormal myelin formation and hind limb weakness Transgenic mice models with overexpression of *Lpin1*: muscle specific overexpression causes increased body weight *Lpin1* ^floxed/floxed^: when deleted in muscle: myopathy	[78,79,80,81]
	*PNPLA2* chromosome 11	ATGL Adipose triglyceride lipase Catalyses the hydrolysis of TAG into DAG	Slowly progressive myopathy: neutral lipid storage disease with myopathy (NLSDM) Arrhythmogenic cardiomyopathy	Distal muscle weakness Cardiomyopathy	Accumulation of lipid droplets(TAG) in multiple tissues Altered energy metabolism Lipidosis of internal organs	*Atgl^−/−^* mice: premature mortality from lipid cardiomyopathy by 16 weeks homozygous knockin mouse model: arrhythmias and significant cardiac dysfunction at 12 weeks, sudden death and/or heart failure by 14 weeks	[82,83,84,85]
	*CHKB* chromosome 22	CHKB Choline kinase β Catalyzes the first step in PE biosynthesis	Megaconial congenital muscular dystrophy	Mitochondrial structural (enlargement) and functional abnormalities	Decreased level of PC	Rostrocaudal muscular dystrophy mice (*rmd*): spontaneous autosomal recessive trait that causes rapidly progressive muscular dystrophy and neonatal forelimb bone deformity	[86,87,88,89]
Disease not listed in ORPHA 352312 despite gene mutation in gene involved in PL biosynthesis, sphingolipids, and FA synthesis	*HACD1* chromosome 10	HACD1 3-Hydroxyacyl-CoA dehydratase 1 Catalyzes the third of the four reactions of the long-chain FA elongation cycle	Congenital fiber-type disproportion, rare internalized nuclei	Congenital myopathy with proximal ± distal muscle weakness, progressively improving over time	Increased concentrations of ≥ C18 and mono-unsaturated FA	Spontaneous *Hacd1* pathogenic variant Labrador retriever dogs: congenital centronuclear myopathy with fiber size disproportion associated with generalized and progressive muscle weakness *Hacd1^−/−^* mice: postural and locomotor weakness and reduced weight gain	[90,91,92]
	*ACADVL* chromosome 17	VLCAD Very long chain Acyl-coA dehydrogenase Involved in the β-oxidation pathway (FA longer than C16)	VLCADD (VLCAD deficiency)	Cardiomyopathy Myopathy with intermittent myalgia and rhabdomyolysis, progressive muscle weakness, often dropped head	Defective long-chain FA oxidation	*Vlcad^−/−^* mice: Cardiac and metabolic dysfunction Different phenotypes between male and female: only KO female mice develop a severe clinical phenotype upon medium chain triglycerides supplementation	[93,94,95,96,97]
	*ACADM* chromosome 1	MCAD Medium-chain acyl-coenzyme A dehydrogenase Involved in the β-oxidation pathway (C4 to C12 acyl chain)	Medium-chain acyl-CoA dehydrogenase deficiency (MCADD)	Myopathy exertion induced myalgia, progressive proximal limb weakness Secondary carnitine deficiency	Accumulation of FA intermediates	*Mcad*^−/−^ mice: organic aciduria and fatty liver, sporadic cardiac lesions, neonatal mortality	[98,99,100]
	*HADHA* chromosome 2	MTP Mitochondrial trifunctional protein Catalyzes three out of the four steps in beta oxidation	Mitochondrial trifunctional protein deficiency (MTP deficiency or MTPD)	Skeletal myopathy, episodic rhabdomyolysis, peripheral neuropathy, cardiomyopathy in severe cases	Decreased long-chain FA oxidation	*Mtpa*^−/−^ mice: severe phenotype (fetal growth retardation, neonatal hypoglycemia) and early sudden death (necrosis and acute degeneration of cardiac and diaphragmatic myocytes)	[101,102,103]
	*CGI-58* chromosome 3	ABHD5 Alpha/beta hydrolase domain containing 5 Lipid droplet-associated protein that activates ATGL	Chanarin–Dorfman syndrome (neutral lipid storage disease)	Moderate myopathy Possible cardiomyopathy	Accumulation of TAG droplets	*Cgi-58* ^−/−^ mice: various phenotypes including cardiomyopathy, severe hepatic steatosis, skin defects	[104,105,106,107,108]
	*CPT2* chromosome 1	CPT2 Carnitine palmitoyltransferase 2 Catalyzes the transfer of FA from cytoplasm to mitochondria	Carnitine palmitoyl-transferase II deficiency	Episodic myalgia, muscle weakness, and rhabdomyolysis with myoglobinuria	Reduced the rates of long-chain FA oxidation into carbon dioxide	Cpt2^floxed/floxed^ mice: cardiac and muscle CPT2 deficiency leads to severe cardiac hypertrophy and ultimately heart failure, with no overt macroscopic muscle phenotype	[109,110,111,112]
Alteration of PL in inherited form of myopathies due to mutation in genes without a direct link to PL biosynthesis	*DYSF* chromosome 2	Dysferlin Involved in the sarcolemma repair mechanism, interacts with caveolin 3	Dysferlinopathy, including: limb girdle muscular dystrophy type 2B (LGMD2B) Miyoshi myopathy	Proximal, distal, or proximo-distal myopathy, predominantly involving lower limbs Rhabdomyolysis	Early intra-myocellular lipid accumulation (sphingolipids, PLs, cholesterol)	various spontaneous murine dysferlin mutants (SJL/J, A/J, BLAJ) with progressive muscular dystrophy	[113,114,115,116]
	*DMD* chromosome X	Dystrophin Involved in the connection of the cytoskeleton of a muscle fiber to the extracellular matrix through the cell membrane	Duchenne muscular dystrophy Becker muscular dystrophy	Cardiomyopathy with myocardial fibrosis, muscle weakness and muscle fiber necrosis, respiratory failure	PL dyshomeostasis Increased PC and SM	Spontaneous pathogenic variant in mice: mdx model Spontaneous pathogenic variant in dog: golden retriever muscular dystrophy Spontaneous pathogenic variant in cat: hypertrophic feline muscular dystrophy Zebrafish model Drosophila model Caenorhabditis model Newly developed rat and pig DMD models	[117,118,119,120,121,122,123]
	*CAV3* chromosome 3	Caveolin3 Component of caveolae plasma membrane (possible scaffolding role for caveolin-interacting molecules)	Rippling muscle disease LGMD Isolated familial hypertrophic cardiomyopathy HyperCKemia and myalgia	Skeletal muscle weakness Muscle hyperirritability triggered by stretch	Decreased CL	*Cav3^−/−^* mice: muscular dystrophy phenotype and alteration of the phenotypic behavior of cardiac myocytes Altered glucose metabolism and increased myoblast proliferation in muscle cells	[124,125,126,127]

## Data Availability

Not applicable.

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
