# Peer review of "Phospholipids: Identification and Implication in Muscle Pathophysiology"

_ijms, 2021, doi:10.3390/ijms22158176_

Round 1

Reviewer 1 Report

In this manuscript, the authors reviewed phospholipids (PL) homeostasis in pathophysiological conditions with particular attention to muscle diseases.

In general, I found the structure of the text flowing and understandable. 

However, I have some minor suggestions that the authors may consider to optimize the manuscript.

Both, the introduction section and paragraph 2 contain too basic information which might be avoided (for instance lanes 38-40; table 1), while more data and findings of PL roles in muscle diseases should be added.  

Author Response

We thank the reviewer for this remark. We agree on the fact that the introduction and paragraph 2 contain basic information. We find important to present basic concepts on phospholipids as some readers may not be familiar with phospholipid characteristics and properties. We have deleted the sentence lanes 38-40, which is quite elementary “For instance, they are able to form micelle spontaneously when they are placed in water. These sphere-shaped structures are possible thanks to their hydrophilic polar head and their hydrophobic tail”. However, we feel that Table 1 is necessary to pinpoint the fact that the position of the double bond is essential, the systematic name provide only the carbon length and the number of unsaturation. If the position of the double bond is known, the common name can be used.

The review contains a large amount of data and findings on the PL roles in muscle diseases. However, in scientific literature, PL have not been widely studied in skeletal muscle. We have extended the data on phospholipids in Sengers Syndrome as the information was incomplete “The depletion by siRNA of AGK in PC-3 prostate cancer cells induce a reduction of LPA and PA in mitochondria (Bektas et al., 2005), whereas in HEK293 (human embryonic kidney) no changes were observed (Vukotic et al., 2017). To the best of our knowledge, no phospholipidomic data are available in muscle cells. “

Reviewer 2 Report

In this very well structured review, Bargui et al. sumarizes the utmost important and updated aspects regarding on current classification, biochemistry, structure, biological functions and pathopysiology of the all known lipids categories related to the biological processes. In fact, I consider of special relevance of this review, as its focusing on a little known topic in scientific literature related to neuromuscular disorders, as the contribution of a lipid disregulation underlying a part of the whole neurmuscular pathophysiology. Interestingly, the review includes a summary and brief description of major techniques employed on lipid analysis, as well as their limitations and recent improvements, exemplified by mass spectrometry (i.e. metabolomis/lipidomics), which has widening the spectrum of lipid types in the biological scenario. I´ve only a minor point to be considered by the authors. According to recent nomenclature in genetics, may be appropriate replace the term "mutation" by "pathogenic variant". In Table 3, the term "recessive mutation" (i.e. in CHKB or HACD1 gene category), may be replaced by "autosomal recessive trait" or "autosomal recessive disorder".

Author Response

We thank the reviewer for this exact remark. We have corrected the text for “pathogenic variant” and “autosomal recessive trait”.

Responses to reviewers

We thank the editor and reviewers for their kind words, constructive comments and suggestions. We have copied the reviewers’ comments, and added our responses below. Changes to the text have been highlighted using the Track Changes function in Word.

Reviewer #2: In this very well structured review, Bargui et al. sumarizes the utmost important and updated aspects regarding on current classification, biochemistry, structure, biological functions and pathopysiology of the all known lipids categories related to the biological processes. In fact, I consider of special relevance of this review, as its focusing on a little known topic in scientific literature related to neuromuscular disorders, as the contribution of a lipid disregulation underlying a part of the whole neurmuscular pathophysiology. Interestingly, the review includes a summary and brief description of major techniques employed on lipid analysis, as well as their limitations and recent improvements, exemplified by mass spectrometry (i.e. metabolomis/lipidomics), which has widening the spectrum of lipid types in the biological scenario. I´ve only a minor point to be considered by the authors. According to recent nomenclature in genetics, may be appropriate replace the term "mutation" by "pathogenic variant". In Table 3, the term "recessive mutation" (i.e. in CHKB or HACD1 gene category), may be replaced by "autosomal recessive trait" or "autosomal recessive disorder".

We thank the reviewer for this exact remark. We have corrected the text for “pathogenic variant” and “autosomal recessive trait”.
